# Direct nucleophilic and electrophilic activation of alcohols using a unified boron-based organocatalyst scaffold

**Jason P. G. Rygus** ◉[1] **& Dennis G. Hall** ◉[1] ✉

Organocatalytic strategies for the direct activation of hydroxy-containing compounds have paled in comparison to those applicable to carbonyl compounds. To this end, boronic acids have emerged as valuable catalysts for the functionalization of hydroxy groups in a mild and selective fashion. Distinct modes of activation in boronic acid-catalyzed transformations are often accomplished by vastly different catalytic species, complicating the design of broadly applicable catalyst classes. Herein, we report the use of benzoxazaborine as a general scaffold for the development of structurally related yet mechanistically divergent catalysts for the direct nucleophilic and electrophilic activation of alcohols under ambient conditions. The utility of these catalysts is demonstrated in the monophosphorylation of vicinal diols and the reductive deoxygenation of benzylic alcohols and ketones respectively. Mechanistic studies of both processes reveal the contrasting nature of key tetravalent boron intermediates in the two catalytic manifolds.

The development of alternative strategies in catalysis is fundamental to the implementation of sustainable chemical processes in organic synthesis[1]. The continued discovery of catalysts capable of activating readily available functional groups, such as alcohols, in an atom-economical fashion under mild conditions represents an evolving frontier in the design of chemical transformations[2]. In this regard, boronic acid catalysis has emerged as a powerful strategy for the atom-economical, metal-free activation of alcohols without stoichiometric derivatization (Fig. 1a)[3,4]. The mild Lewis acidity of boronic acids, in conjunction with their ability to undergo reversible covalent exchange with hydroxy-containing substrates, can activate a wide variety of alcohols towards subsequent transformations under mild and selective conditions[3].

The mechanisms of activation in boronic acid catalysis are decidedly substrate and catalyst dependent. Highly electron-deficient catalysts (particularly cationic or heavily fluorinated arylboronic acids) that exhibit increased acidity can provide electrophilic activation of an alcohol towards nucleophilic substitution, often by an $S_N1$ mechanism (Fig. 1b)[5–12]. In contrast, nucleophilic activation of polyol substrates generally proceeds through formation of an anionic tetravalent adduct

that displays oxygen-centered nucleophilicity (Fig. 1c)[13]. These adducts are formed readily when oxidatively-sensitive borinic acids are employed as catalysts[14] due to their single exchangeable boranol (B–OH) unit, whereas an additional Lewis base is required to generate the analogous tetravalent adduct from a boronic acid[15]. Our laboratory has recently reported the use of BINOL-derived cyclic hemiboronic acids, which combine the oxidative stability of a boronic acid with the single exchangeable site of a borinic acid, as highly effective catalysts for the enantioselective desymmetrization of 1,3-diols via O-benzylation through a tetravalent dialkoxyboronate anion intermediate[16].

The multitude of transformations that have proven amenable to boronic acid catalysis highlights how distinct modes of catalytic activation are largely accomplished by remarkably different boron species. In the absence of privileged catalyst scaffolds[17], reaction discovery often necessitates substantial catalyst screening[18]. Accordingly, the development of a universal catalyst framework represents an alluring prospect in catalysis, where divergent catalytic applications can be guided by fundamental catalyst reactivity. A rigorous understanding of the underlying properties of boronic acid catalysts (such as their exchangeability with nucleophiles, their acidity, and their stability) is

[1]Department of Chemistry, Centennial Center for Interdisciplinary Science, University of Alberta, Edmonton, AB T6G 2G2, Canada.
✉e-mail: dennis.hall@ualberta.ca

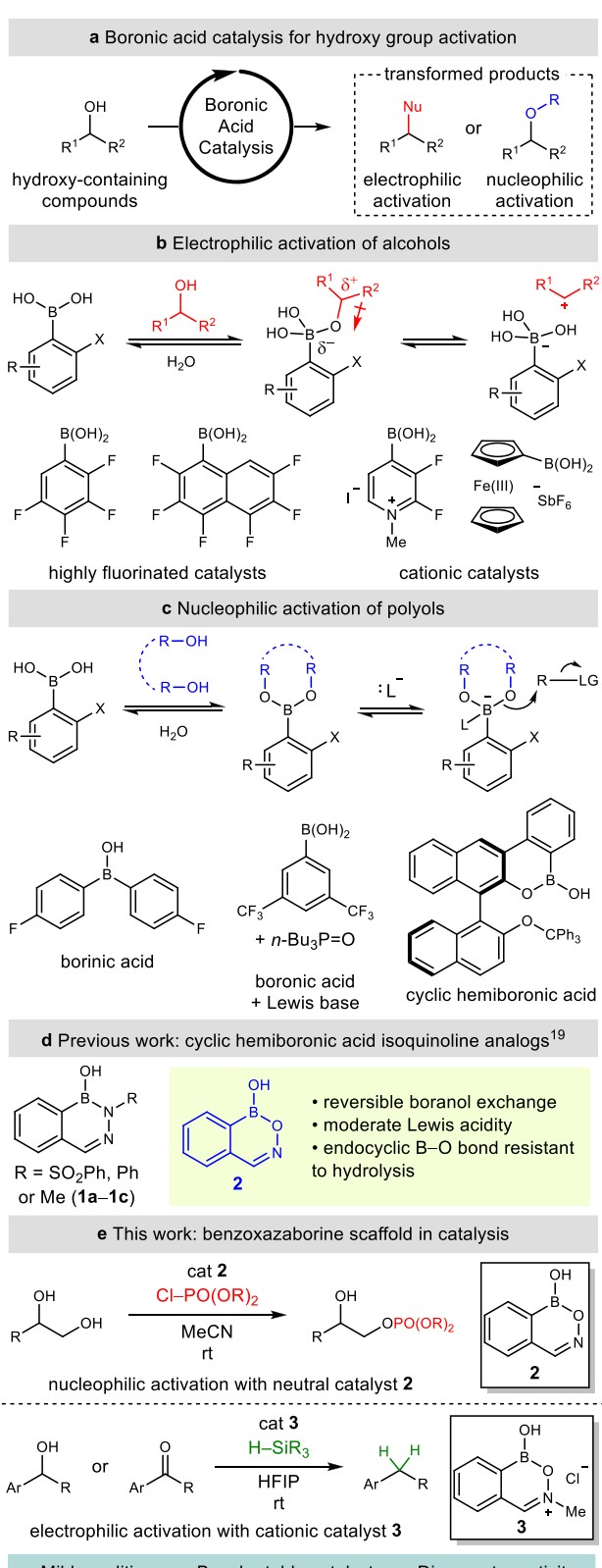

**Fig. 1 | Boronic acid catalysis for hydroxy group activation. a** Concept of boronic acid catalysis. **b** Boronic acid-catalyzed electrophilic activation of alcohols. **c** Boronic acid-catalyzed nucleophilic activation of diols. **d** Cyclic hemiboronic acid isoquinoline analogs. **e** Mechanistically divergent catalytic applications of the benzoxazaborine scaffold (this work).

essential toward building a mechanistic foundation upon which to develop a privileged catalyst scaffold.

Along these lines, our laboratory has recently reported a comprehensive study on the acidity and intrinsic reactivity of boranol-containing naphthoid heterocycles (Fig. 1d)[19]. While their Lewis acidic character was unambiguously established, the strength of their acidity and exchangeability of their boranol hydroxy group varied greatly. The benzoxazaborine (**2**) scaffold demonstrated several properties ideal for the development of a universal, mechanistically divergent catalyst scaffold—including rapid covalent exchange, moderate Lewis acidity, and a resistance towards endocyclic B–O hydrolysis[19]. We envisioned that while the moderate p$K_a$ of benzoxazaborine **2** could be ideal for nucleophilic catalysis, the design of a highly acidic analog of this scaffold for electrophilic catalysis could enable a mechanistically divergent approach to the functionalization of alcohols from a common heterocyclic framework.

Herein we report the successful application of this strategy using boron heterocycles **2** and **3** to catalyze nucleophilic and electrophilic activation of diols and alcohols, respectively. This divergent reactivity is exemplified in the selective monophosphorylation of vicinal diols and the reductive deoxygenation of π-activated alcohols and ketones (Fig. 1e). These reactions proceed under ambient conditions using easily synthesized bench-stable catalysts and demonstrate a clear association between the application of boron-based catalysts and their underlying fundamental properties and reactivity.

## Results

### Monophosphorylation of vicinal diols

The selective phosphorylation of polyhydroxylated compounds is an essential biosynthetic process in kinase-mediated ATP- and GTP-dependent signaling pathways[20]. In drug discovery, phosphorylation can be used as a prodrug strategy to enhance water-solubility upon hydrolysis[21], while in vivo monophosphorylation of a 1,3-diol moiety in immunosuppressive compound FTY720 is essential to its biological activity[22]. Catalytic monophosphorylation in chemical synthesis has largely been limited to 1,3-diol substrates by employing Lewis acidic titanium[23] or silver-based catalysts[24], including enantioselective desymmetrization approaches[25,26]. In comparison to the use of inorganic Lewis acid catalysts, hemiboronic acid-catalyzed monophosphorylation may employ a benign organocatalyst that is effective under mild reaction conditions. We envisioned that a nucleophilic tetravalent dialkoxyboronate anion (cf. Fig. 1c) could be accessed from benzoxazaborine **2** and vicinal diols under mild conditions. Provided the uncatalyzed background reaction is sufficiently slow, the enhancement of nucleophilicity afforded by the chelated boronate should provide high selectivity for monofunctionalization upon electrophilic trapping. Accordingly, the ability of benzoxazaborine **2** to promote monophosphorylation of vicinal diols using a chlorophosphate electrophile was investigated.

We first examined stoichiometric reactivity between benzoxazaborine **2** and vicinal diol **4a** using $^{11}$B NMR spectroscopy (Fig. 2a). Covalent boranol exchange was observed rapidly at room temperature in $d_3$-acetonitrile consistent with equilibrium generation of hemiboronic ester **2-I**. Reaction with a weak base (N,N-diisopropylethylamine, DIPEA) led to virtually quantitative formation of the corresponding tetravalent boronate **2-II**, which displayed a characteristic upfield $^{11}$B NMR resonance (7.2 ppm). Subsequent $^1$H NMR studies suggested that due to stereochemistry at the tetravalent boron atom, boronate **2-II** exists as a mixture of two diastereomers, where the rate of interconversion is dependent on the nature of the base (Supplementary Figs. 5 and 6). Upon addition of diethyl chlorophosphate, boronate **2-II** was rapidly quenched to restore free hemiboronic acid **2** and generate monophosphorylated alcohol **5a**. Each of these elementary steps occurred rapidly (<5 min) at room temperature with no exclusion of air or moisture. Catalytic monophosphorylation was

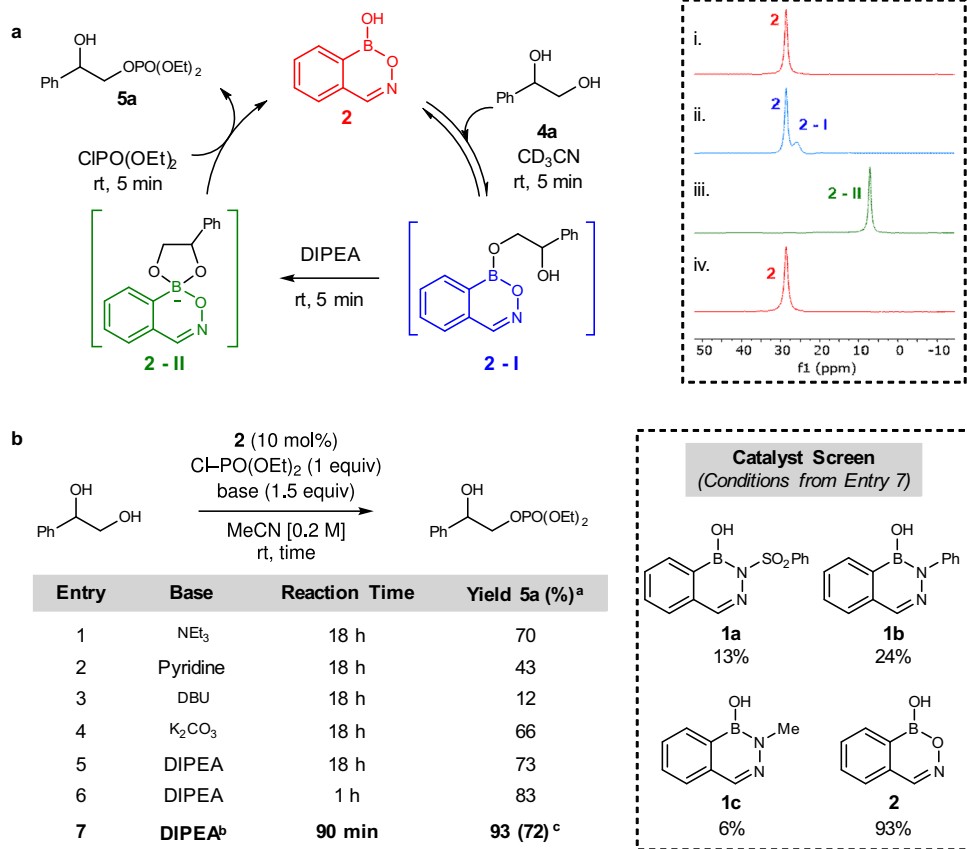

**Fig. 2 | Monophosphorylation of vicinal diols. a** Initial stoichiometric reactivity between boron heterocycle **2** and diol **4a**. **b** Selected optimization of catalytic monophosphorylation reaction. [a]Yields determined by $^1$H NMR relative to 1,3,5-trimethoxybenzene as an internal standard. [b]1.1 equivalents of DIPEA and ClPO(OEt)$_2$. [c]Isolated yield. In all cases, rr > 98:2.

subsequently optimized and found to proceed effectively with DIPEA as a base in only 90 min (Fig. 2b). In line with a sterically preferred attack of the least hindered oxygen atom of borate complex **2-II**, phosphorylation occurred with complete regioselectivity for the primary alcohol in all cases. Control reactions revealed only trace product formation in the uncatalyzed background reaction, and significantly reduced yield for a primary alcohol lacking the vicinal diol moiety (see Supplementary Information Section 4.3). Benzoxazaborine **2** demonstrated significantly improved catalytic activity in this transformation relative to its aza-congeners **1a**–**1c**[19].

In stoichiometric experiments, conversion of heterocycles **1**–**2** to the corresponding tetravalent adducts **II** was strongly correlated to the acidity of the parent hemiboronic acid (Fig. 3a). However, adducts derived from strongly Lewis acidic heterocycles demonstrate diminished nucleophilicity in subsequent electrophile trapping (see Supplementary Information Section 4.4). Thus, the unique effectiveness of heterocycle **2** as a catalyst appears to originate from an appropriate balancing act of Lewis acidity-driven conversion to a tetravalent boronate adduct with sufficient nucleophilicity[20]. Finally, the substrate scope of the reaction was examined with respect to the 1,2-diol component **4**, where a variety of 1-aryl substituted 1,2-ethanediols underwent regioselective phosphorylation in moderate to good yield (Fig. 3b).

## Reductive deoxygenation

A moderate p$K_a$ is generally desired for boron-based catalysts in nucleophilic diol activation so that an appropriate balancing can be achieved of effective substrate binding without attenuated nucleophilicity of the catalyst-substrate complex. In contrast, electrophilic activation of alcohols ultimately involves partial or complete

ionization of the C−O bond to form a carbocation intermediate whose lifetime is inversely correlated to the nucleophilicity of the associated hydroxyboronate anion[5]. While multiple mechanisms of activation may be operative in these processes[10]−including hydrogen bond activation or Lewis acid-assisted Brønsted acidity – catalytic efficiency of boronic acids in electrophilic activation is often correlated to their acidity wherein catalysts with lower p$K_a$ generate a more stable hydroxyboronate anion upon C−O activation[11]. Numerous strategies have been demonstrated to lower the p$K_a$ of a boronic acid, including the introduction of fluorine substituents[27−29] and intramolecular hydrogen bonding in *ortho*-substituted arylboronates[30]. An alternate strategy to increase catalytic efficiency in electrophilic activation is the use of cationic boronic acids. Upon alcohol activation, ion exchange of the resulting zwitterionic hydroxyboronate can afford a reactive carbocation and impede C−O recombination. Our laboratory has previously demonstrated that ferrocenium boronic acid hexafluoroantimonate salt is an exceedingly active catalyst relative to the parent ferrocene analog in Friedel-Crafts benzylation with deactivated alcohols[5].

Through our design of a cationic analog of benzoxazaborine **2**, we found that condensation of commercially available 2-formylphenylboronic acid with N-methylhydroxylamine hydrochloride under ambient conditions readily afforded benzoxazaborinium salt **3** on gram-scale, which proved stable to storage under air with no exclusion of moisture (Fig. 4a)[31]. Connectivity of the iminium moiety was unambiguously established with an X-ray crystallographic structure of the corresponding tetravalentbis(hexafluoroisopropoxy)boronate zwitterion **3-II** (Fig. 4b)[32]. The p$K_a$'s of heterocycles **2** and **3** were measured by $^{11}$B NMR titration in D$_2$O, where iminium **3** displayed a tetravalent boron as low as pH 0.8, corresponding to a p$K_a$ less than 1

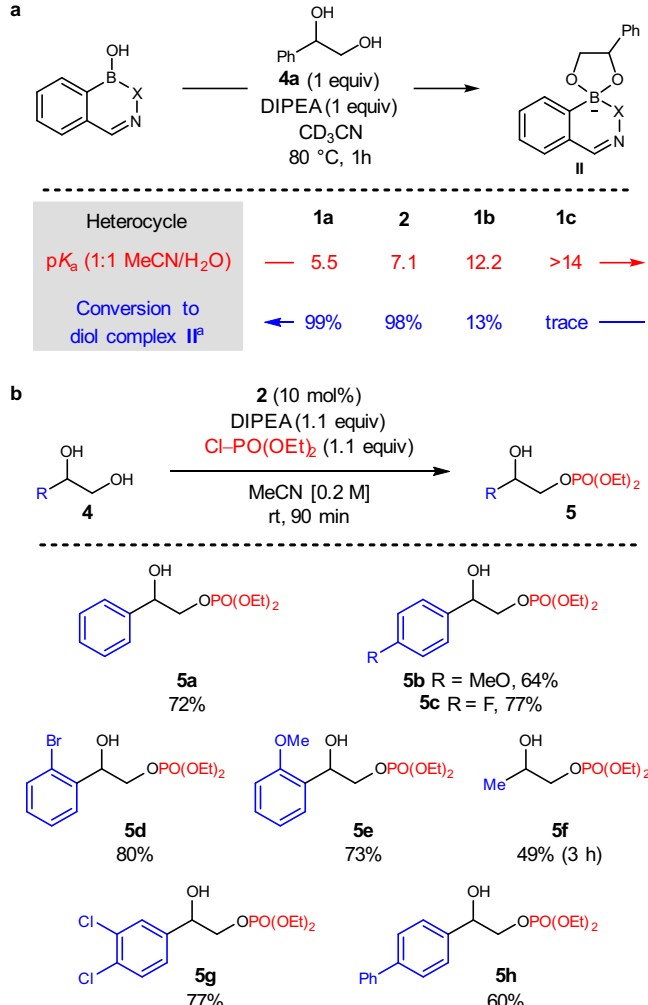

**Fig. 3 | Hemiboronic acid-catalyzed monophosphorylation of vicinal diols.**
**a** Inverse correlation between hemiboronic acid $pK_a$ and conversion to the corresponding tetravalent diol complex. [a]Determined by integration of $^{11}B$ NMR resonances. For full details, see Supplementary Information Section 4.4. **b** Scope of the monophosphorylation reaction catalyzed by heterocycle **2**.

and a minimum 30,000-fold increase in acidity relative to benzoxazaborine **2** ($pK_a$ 5.5)[19].

To assess the activity of heterocycle **3** in the electrophilic activation of alcohols, we were compelled to examine the reductive deoxygenation of carbon-oxygen bonds. Defunctionalization strategies[33] have significant utility in the late-stage modification of bioactive molecules[34] or the reduction of lignin and other biomass-derived feedstocks[35,36]. Deoxygenation reactions offer an indispensable approach for converting abundant, readily accessible oxygenated building blocks such as alcohols and ketones into less densely functionalized species. Traditionally, alcohol deoxygenation can be accomplished by means of the Barton–McCombie reaction using toxic tin hydride reagents after stoichiometric xanthone formation[37,38], while ketones can be converted to the corresponding methylene unit via a hydrazone intermediate through the Wolff–Kishner reduction at high temperature under strongly basic conditions[39]. Activated C–O bonds are also susceptible to hydrogenolysis in the presence of transition metal catalysts, although dehalogenation of aryl halide substituents can limit the applicability of these methods[40,41]. Catalytic deoxygenation strategies using silanes as benign hydride donors can offer improved functional group tolerance and atom economy without requiring stoichiometric pre-activation. While a variety of metal salts

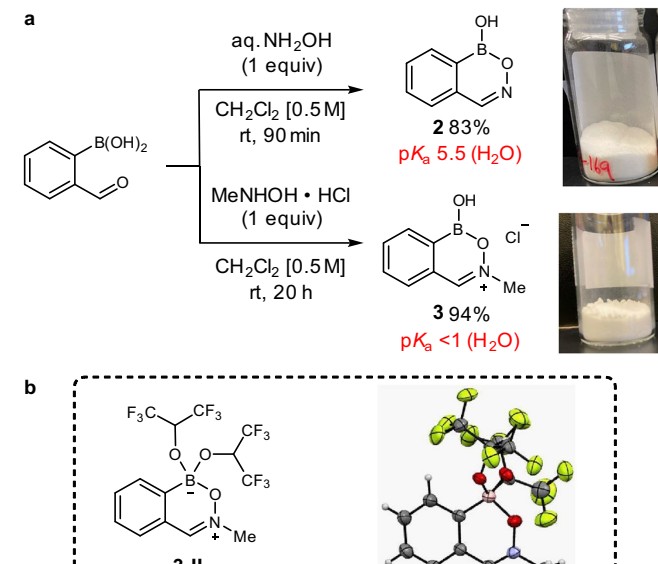

**Fig. 4 | Catalyst synthesis and characterization. a** Synthesis of boron heterocycles **2** and **3**. **b** ORTEP representation of zwitterionic bis(hexafluoroisopropoxy) boronate **3-II**.

### Table 1 | Optimization of deoxygenation reaction conditions

| Entry | Mol% Catalyst | Solvent | Concentration | Yield 7a (%)[a] |
|---|---|---|---|---|
| 1 | 10 mol% | HFIP/MeNO₂ 4:1 | 0.5 M | 97% |
| 2 | 1 mol% | HFIP/MeNO₂ 4:1 | 0.5 M | 98% |
| 3 | 1 mol% | HFIP/MeNO₂ 1:4 | 0.5 M | 68% |
| 4 | 1 mol% | MeNO₂ | 0.5 M | 0% |
| 5 | 1 mol% | MeCN | 0.5 M | 0% |
| 6 | 1 mol% | TFE/MeNO₂ 4:1 | 0.5 M | 38% |
| 7 | 1 mol% | HFIP/MeNO₂ 4:1 | 2.0 M | 98% |
| 8 | 1 mol% | HFIP | 2.0 M | 90% |

[a]Yields determined by $^{1}H$ NMR relative to 1,3,5-trimethoxybenzene as an internal standard.
TFE = 2,2,2-trifluoroethanol.

have been shown to promote silane-mediated reductive deoxygenation[42,43], the emergence of boron-based catalysts for this transformation are particularly attractive due to their tunable Lewis acidity and benign environmental impact[44]. The highly Lewis acidic $B(C_6F_5)_3$ has demonstrated exceptional catalytic activity in the deoxygenation of alcohols[45], ethers[46], aldehydes[47], ketones[48] and amides[49], while other air- and moisture-sensitive fluorinated arylborane catalysts have been reported[50]. The development of a bench-stable boron-based catalyst for these processes is highly desirable to afford practical deoxygenation reactions under mild conditions[51]. We hypothesized that reaction of a π-activated alcohol with highly acidic catalyst **3** should lead to productive C–O ionization, and subsequent trapping of the carbocation by silane could furnish the reduction product with concomitant Si–O bond formation to regenerate the catalyst.

The reductive deoxygenation of diphenylmethanol **6a** to afford diphenylmethane **7a** was examined in initial optimization using triethylsilane as a reducing agent and heterocycle **3** as a catalyst (Table 1). Based on related reports from our laboratory[5,52], mixtures of 1,1,1,3,3,3-hexafluoroisopropanol (HFIP) and nitromethane were initially

**Fig. 5 | Substrate scope for the reductive deoxygenation of alcohols.** Scope of di- and triarylalkane products obtained via reductive deoxygenation. Reaction conditions: **6** (1.0 equiv), **3** (0.1–5 mol%), HSiEt$_3$ (1.1 equiv), HFIP/MeNO$_2$ (4:1, 2.0 M in **6**), room temperature, 1.5–18 h.

examined as solvents for this transformation. HFIP demonstrates enhanced acidity and hydrogen bond donor ability relative to aliphatic alcohols and offers a suitably high dielectric constant with low nucleophilicity to effectively stabilize cationic intermediates[53], while nitromethane promotes solubility and has previously been reported to act as a hydrogen bond acceptor in acid-catalyzed reactions[54]. Full conversion to the desired product was observed in 90 min at room temperature using 4:1 HFIP/MeNO$_2$ as the solvent mixture with catalyst loadings down to 1 mol%. The reaction could be conducted at concentrations of 2.0 M with no loss of catalytic activity, providing significant solvent economy (see Supplementary Information Section 7.1 for full optimization).

The deoxygenation conditions proved applicable to a wide scope of substituents (Fig. 5). Halide-substituted aromatic rings were well tolerated (**7b**–**7g**) with no evidence for hydrodehalogenation which may occur in transition metal catalyzed processes. Chemoselective reduction of a secondary diarylmethanol moiety was accomplished

with no reduction of a primary benzylic alcohol to afford **7h**. As a testament to the mildness of this method, catalyst **3** showed improved selectivity in the deoxygenation of alcohol **6h** relative to traditional Lewis or Brønsted acids (see Supplementary Information Section 8.3). A sterically hindered 2,6-dimethylsubstituted product (**7k**) was formed in good yield, while a methyl benzoate ester showed no competing ester reduction in the formation of **7n**. A heteroaromatic thiophene-substituted analog (**7o**) was prepared in good yield, and an internal allylic alcohol was successfully deoxygenated without reduction of the conjugated alkene to afford styrene derivative **7p**. Reduction of triphenylmethanol to afford triphenylmethane **7r** proceeded in high yield on multigram-scale with reduced catalyst loading of 0.1 mol%. Diarylethane derivative **7s**, an antagonist for the smallpox virulence factor N1L protein[55], was successfully prepared by reduction of the corresponding tertiary alcohol. Reduction of cardiovascular drug cloridarol[56] proceeded smoothly to afford benzofuran-substituted diarylmethane **7t**. Furthermore, secondary acetophenone-derived

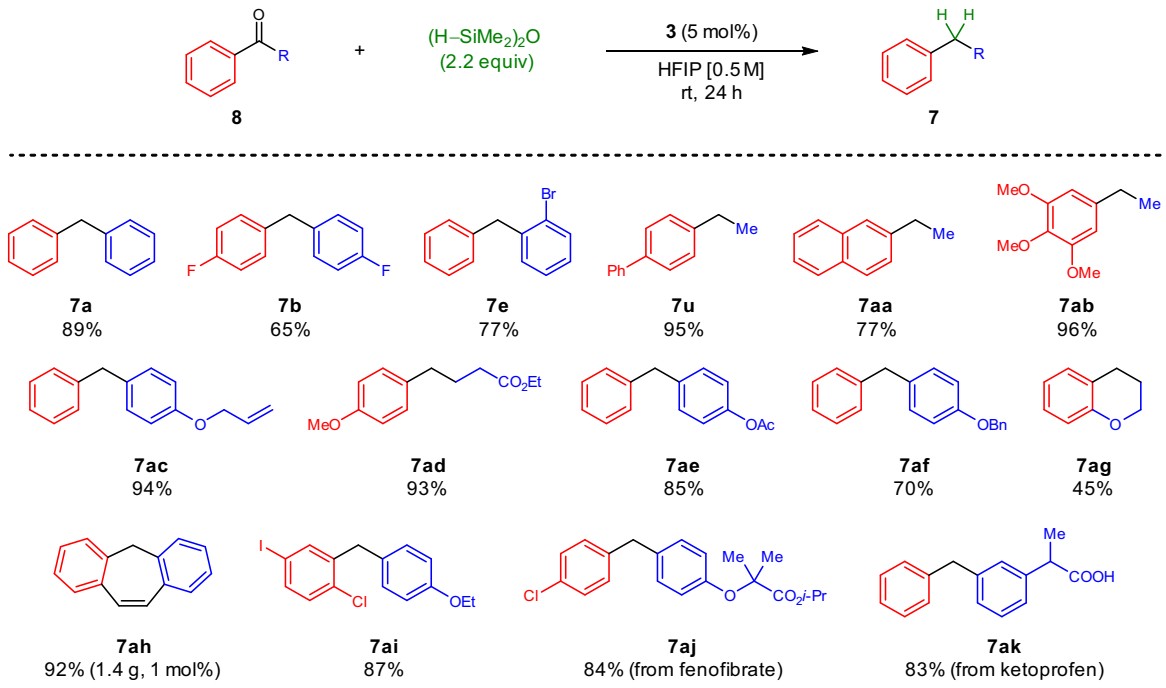

**Fig. 6 | Substrate scope for the reductive deoxygenation of ketones.** Scope of reduction products obtained via reductive deoxygenation of acetophenone or benzophenone derivatives. Reaction conditions: **8** (1.0 equiv), (HSiMe$_2$)$_2$O (2.2 equiv), **3** (5 mol%), HFIP (0.5 M in **8**), room temperature, 24 h.

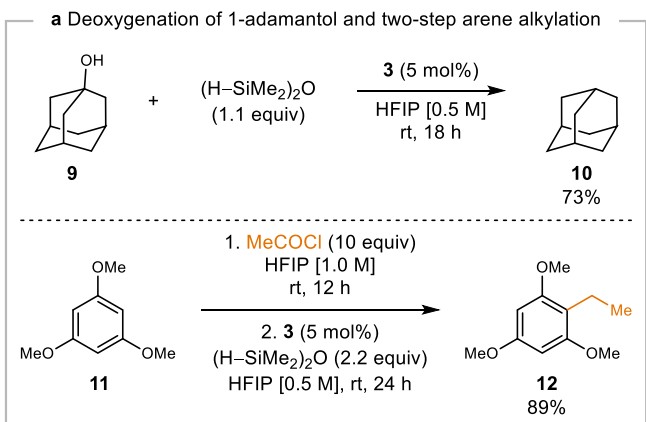

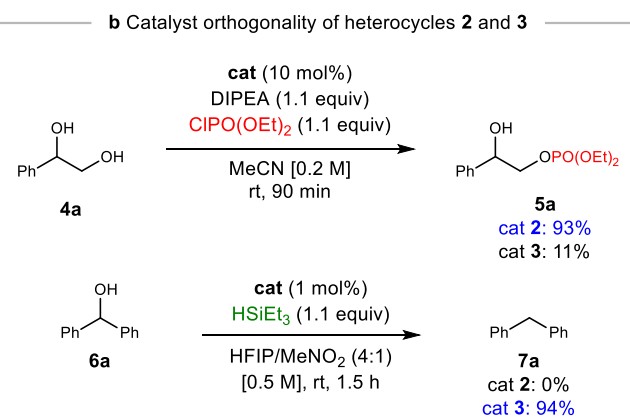

**Fig. 7 | Synthetic extensions of catalytic reductive deoxygenation and catalyst orthogonality. a** Reductive deoxygenation of 1-adamantol and two-step ethylation of 1,3,5-trimethoxybenzene at room temperature. **b** Comparison of catalysts **2** and **3** in model phosphorylation and deoxygenation reactions.

alcohols with only a single activating π-system were viable deoxygenation substrates under slightly modified conditions. Synthesis of **7x** was accomplished with no competing alkene reduction, and an α-cyclopropyl substituent was well tolerated (**7y**). Other substrates with highly electron-deficient aromatic groups were ineffective in this methodology (see Supplementary Information Section 8.4 for a list of failed substrates).

Remarkably, the deoxygenation protocol could further be extended to aromatic ketones under modified conditions (Fig. 6). The use of 1,1,3,3-tetramethyldisiloxane (TMDSO)[57] as a hydride source was essential to this process and gave significantly improved yields compared to triethylsilane (see Supplementary Information Section 7.2 for full optimization). Ketone deoxygenation catalyzed by heterocycle **3** proceeds under ambient conditions with no exclusion of air or moisture and displayed comparable functional group tolerance to the previously described alcohol deoxygenation protocol. Deoxygenation of dibenzosuberenone to afford **7ah** proceeded in high yield on gram-scale with reduced catalyst loading. Remarkably, dihalide-functionalized diarylmethane **7ai**–an intermediate in the synthesis of sotagliflozin, an SGLT-1/2 inhibitor used in the treatment of diabetes[58]–was prepared from the corresponding ketone in significantly improved yield compared to the reported alternative using excess boron trifluoride diethyl etherate[59]. Furthermore, chemoselective late-stage deoxygenation of benzophenone-containing bioactive compounds fenofibrate and ketoprofen proceeded effectively to afford diarylmethanes **7aj** and **7ak**, respectively, without additional undesired C=O or C–O bond reductions.

Alcohol deoxygenation could be extended beyond π-activated alcohols to the reduction of tertiary alcohol 1-adamantol (**9**), affording adamantane (**10**) in good yield (Fig. 7a). A two-step process involving HFIP-mediated acylation[60] of trimethoxybenzene **11** and subsequent ketone deoxygenation with catalyst **3** was found to generate arene **12** in good yield in a formal Friedel-Crafts primary alkylation (Fig. 7a). In contrast to previous syntheses of **12** from **11**[61,62], the expedient synthesis reported herein occurs at room temperature under ambient conditions with only a single purification.

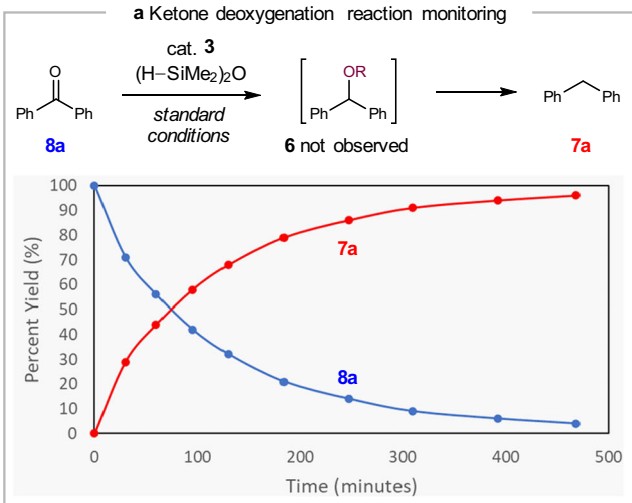

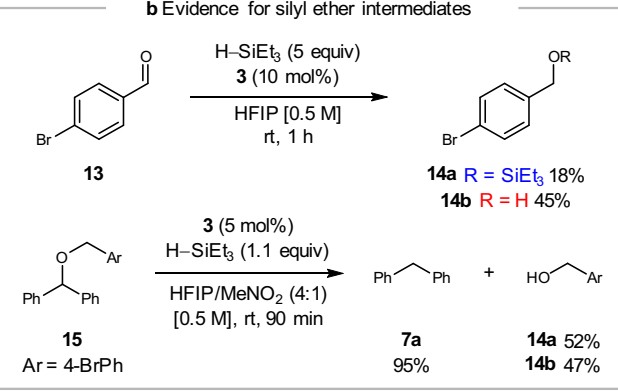

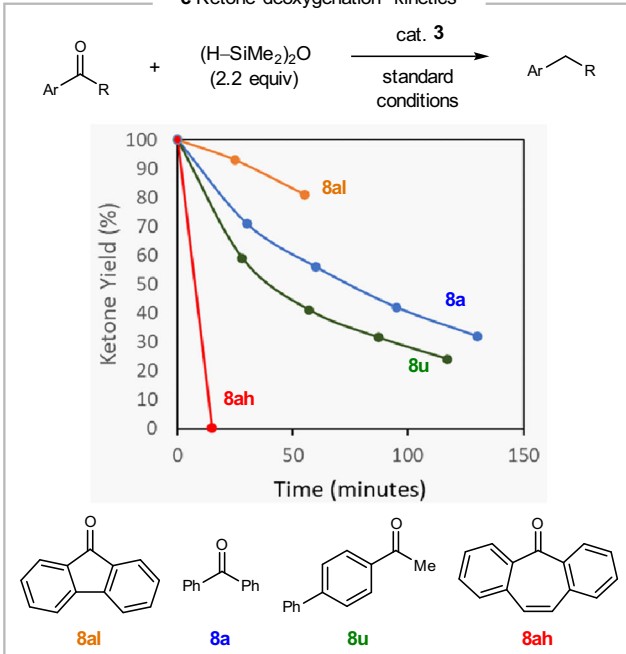

**Fig. 8 | Further investigation of ketone reductive deoxygenation. a** Reaction profile for the deoxygenation of ketone **8a. b** Evidence for the formation and hydrolysis of an intermediate silyl ether in carbonyl deoxygenation. **c** Effect of substrate electronics on the rate of ketone deoxygenation.

Despite the structural similarity of catalysts **2** and **3**, there is significant mechanistic divergence in their modes of activation. Iminium catalyst **3** showed a nearly ten-fold reduction in activity for phosphorylation relative to neutral heterocycle **2**, reflecting the dramatic reduction in nucleophilicity of a zwitterionic tetravalent diol adduct formed from **3** relative to an anionic boronate derived from **2**. In contrast, benzoxazaborine **2** was entirely inactive as a catalyst for alcohol deoxygenation, highlighting the dramatic enhancement in electrophilic activation observed with catalyst **3** (Fig. 7b).

Additional mechanistic studies were conducted to further probe ketone deoxygenation catalyzed by heterocycle **3**. We hypothesized that deoxygenation of benzophenone **8a** may proceed through an initial hydrosilylation to afford the corresponding secondary silyl ether **6aSi**, or alcohol **6a** upon in situ alcoholysis (Fig. 8a). Subsequent C–O bond ionization and trapping of the resulting carbocation intermediate though hydride transfer from the silane would afford the reduction product **7a**. When the conversion of **8a** to **7a** was monitored by [1]H NMR, no evidence for intermediates **6a** or **6aSi** was observed. Upon subjecting 4-bromobenzaldehyde **13** to modified deoxygenation conditions using triethylsilane, a mixture of triethylsilyl ether **14a** and benzyl alcohol **14b** was observed (Fig. 8b). Silyl ether **14a** showed only trace desilylation in HFIP alone, but significant conversion to alcohol **14b** was observed upon reaction with catalyst **3** in HFIP. Furthermore, reduction of mixed benzhydryl benzyl ether **15** occurred chemoselectively at the secondary C–O bond, liberating the primary benzyl alcohol as a mixture of silyl ether **14a** and free alcohol **14b**. These results are consistent with the formation of silyl ether intermediates in ketone deoxygenation reactions catalyzed by heterocycle **3**.

Ketone deoxygenation was found to proceed more rapidly for acetophenone derivative **8u** than benzophenone **8a**, consistent with an increasingly electrophilic carbon center (Fig. 8c). It is noteworthy that this trend is opposite for the reactivity of the corresponding alcohols **6u** and **6a**, for which reduction of diphenylmethanol **6a** proceeds faster and with lower catalyst loading (cf. Fig. 5). This observation suggests that stability of a putative carbocation intermediate is far more influential on the rate of C–O bond activation than on carbonyl hydrosilylation. Furthermore, the rate of ketone deoxygenation was highly sensitive to central-ring aromaticity in fused diarylketones. Reduction of 9-fluorenone **8al**, for which the intermediate cation displays anti-aromaticity[63], proceeds significantly slower than benzophenone **8a**. In contrast, deoxygenation of 5-dibenzosuberenone **8ah**, in which the central ring contains 6 π-electrons, is complete within 15 min under the standard conditions.

When heterocycle **3** was dissolved in HFIP, a broad resonance at 26.2 ppm was observed by [11]B NMR spectroscopy, corresponding to a trivalent compound which was suggested by ESI analysis (positive mode) to be the hexafluoroisopropoxy ester formed through boranol exchange with solvent (Fig. 9). Upon addition of substrate **6a**, clean conversion to a tetravalent boron environment was observed (6.0 ppm), consistent with C–O ionization. Subsequent treatment with silane restored a trivalent boron compound (24.8 ppm), and afforded reduction product **7a** as observed by [1]H NMR spectroscopy. Furthermore, tetravalent zwitterionic bis(hexafluoroisopropoxy)boronate **3-II** (cf. Fig. 4b) showed no catalytic activity in both alcohol and ketone deoxygenation. This is further consistent with a trivalent boron species as the active catalyst and suggests that no pre-equilibration occurs between trivalent and tetravalent boron species prior to substrate activation. In contrast to carbonyl hydrosilylations promoted by B(C6F5)3, no evidence was observed for borohydride formation through Si–H activation of TMDSO by heterocycle **3** alone in the absence of substrate[64]. Further studies are ongoing in our laboratory to elucidate the mechanism of C–O ionization and ketone activation, which may involve a combination of Lewis acid, Brønsted acid or silylium ion catalysis[65,66].

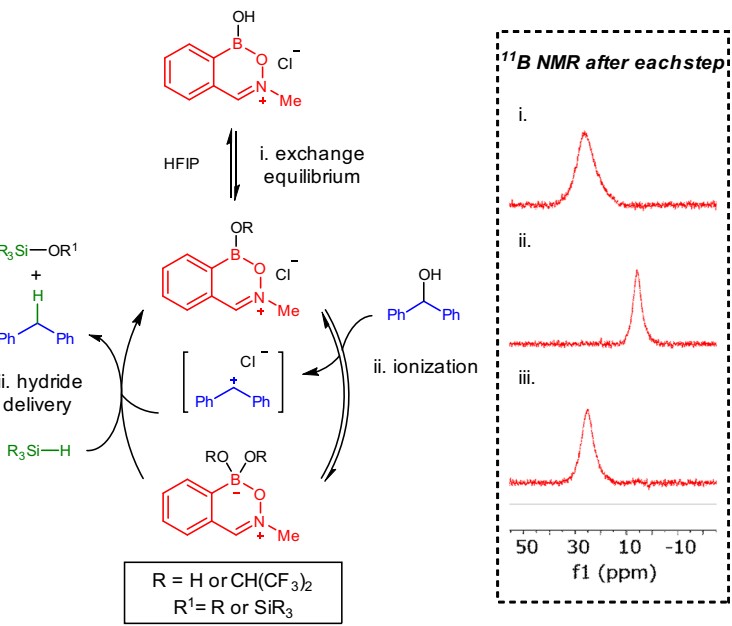

**Fig. 9 | Proposed catalytic cycle for alcohol deoxygenation.** Proposed reaction pathway for alcohol deoxygenation supported by $^{11}$B NMR experiments.

In summary, the benzoxazaborine scaffold offers a versatile organocatalytic platform for the development of hemiboronic acid catalysts for the direct activation and functionalization of hydroxy-containing compounds under ambient conditions. The parent neutral heterocycle is an effective catalyst for the monophosphorylation of vicinal diols, while a structurally related cationic catalyst is highly active for the reductive deoxygenation of alcohols and ketones with silanes. Mechanistic studies of both processes revealed the essential interplay of trivalent and tetravalent intermediates during catalysis stemming from the subtle yet profound structural differences between catalysts **2** and **3**. These results demonstrate a clear link between the fundamental properties of boron-containing heterocycles and their catalytic activity. With strategic modifications to established scaffolds, we anticipate that the results described herein constitute an attractive starting point for the rational design and development of hemiboronic acid catalysts and boronic acid-catalyzed transformations.

## Methods
### General procedure for monophosphorylation of vicinal diols
A two-dram vial with a stir bar was charged with diol **4** (1.0 equiv), catalyst **2** (10 mol%) and MeCN (0.2 M). The reaction was stirred for 30 s until fully dissolved, followed by addition of DIPEA (1.1 equiv) and ClPO(OEt)$_2$ (1.1 equiv) (caution: addition of the electrophile is mildly exothermic). The vial was capped and stirred at room temperature for 1.5 h. Upon completion, the reaction mixture was diluted with ethyl acetate (20 mL) and washed successively with 1 M HCl$_{(aq)}$ (10 mL), saturated NaHCO$_{3(aq)}$ (10 mL) and brine (10 mL). The organic layer was dried over Na$_2$SO$_4$, filtered, and concentrated by rotary evaporation. Purification by column chromatography afforded the desired product **5**.

### General procedure for reductive deoxygenation of alcohols
A vial equipped with a stir bar was charged with alcohol **6**, catalyst **3** (0.1–5 mol%), triethylsilane (1.1 equiv), HFIP and MeNO$_2$ (4:1 ratio, 2.0 M in alcohol **6**). The reaction was stirred at room temperature for the indicated reaction time, after which it was concentrated by rotary evaporation. Purification by column chromatography afforded the reduction product **7**.

### General procedure for reductive deoxygenation of ketones
A vial equipped with a stir bar was charged with ketone **8**, catalyst **3** (5 mol%), 1,1,3,3-tetramethyldisiloxane (TMDSO) (2.2 equiv) and HFIP (0.5 M in ketone **8**). The reaction was stirred at room temperature for 24 h, after which it was concentrated by rotary evaporation. Purification by column chromatography afforded the reduction product **7**.

## Data availability
The crystallographic data for compound **3-II** has been deposited in the Cambridge Crystallographic Data Center (CCDC) under deposition number CCDC 2210073. Copies of the data can be obtained free of charge via https://www.ccdc.cam.ac.uk/structures/. All other data supporting this study are available in the Supplementary Information, or from the corresponding author upon request.

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

## Acknowledgements

This work was supported by the Natural Science and Engineering Research Council of Canada (NSERC, grant RGPIN-2017-05086 for D.G.H.), the Canada Research Chairs Program and the University of Alberta. J.P.G.R. is thankful to NSERC and the Province of Alberta for graduate funding. The authors wish to thank Dr. Michael J. Ferguson (X-ray Crystallography Laboratory, University of Alberta) for the X-ray crystallographic analysis of compound 3-II.

## Author contributions

J.P.G.R. and D.G.H. conceived the study. J.P.G.R. performed the experiments. The manuscript and Supplementary Information were written by J.P.G.R. with assistance from D.G.H.

## Competing interests

The authors declare no competing interests.
