## [Peer Review File · Nature Communications]

REVIEWER COMMENTS

Reviewer #1 (Remarks to the Author):

This manuscript reports the exploration of benzoxazaborines for the catalytic activation of alcohols. This work is unique in that the same parent scaffold can be used for the activation of alcohols towards reaction with electrophiles and nucleophiles, which have very different requirements of the catalyst. The parent benzoxazaborine catalyst enhances the nucleophilicity of diols, which is exemplified by a mono-phosphorylation reaction. Formation of the N-methylated benzoxazaborinium analogue changes the nature of the catalyst, allowing activation of alcohols and ketones towards reduction into the corresponding alkane using alkylsilanes. Mechanistic control experiments are provided for both sets of reactions to provide supporting evidence for the differing roles of the catalysts in each process. The substrate scope and limitations are also discussed for both reaction classes. The manuscript is prepared to a very high standard and was a pleasure to read. The Supporting Information is detailed, the products well characterised, and the NMR spectra show sufficient purity.

While the individual reactions reported have some precedence in arylboronic / borinic acid catalysis, the breakthrough in this work is the development and demonstrated versatility of the benzoxazaborine scaffold for catalysis. The mechanistic insights shown ensures that there is great potential for further exploration of these interesting catalysts that I am sure will be of wide interest to the community. Publication in Nature Communications can therefore be highly recommended, essentially without correction.

Minor Comment

SI Page 6 – The graph in Figure S1 is of low resolution compared with the other figures. It is still readable, but the authors may want to update.

Dr James Taylor

University of Bath

Reviewer #2 (Remarks to the Author):

Hall et al. developed boronic acid catalyst for monophosphorylation of vicinal diols and boronic acid catalyst for reductive deoxygenation of alcohols and ketones. These two boronic acid catalysts were designed based on a common benzoxazirine scaffold. However, these simple two reactions have been already known. The novelty and substrate scope are not acceptable as a paper in Nat. Commun.

Monophosphorylation of vicinal diols: In general, it is well known that primary hydroxy group more reactive than secondary hydroxy group. In particular, secondary hydroxy group of 1-aryl-1,2-ethanediol is much less reactive than primary hydroxy group. The regioselectivity should be shown in Figure 2. The substrate scope is quite limited.

Reductive deoxygenation of alcohols and ketones: In general, diarylmethanol and diarylketone are highly reactive for the reductive deoxygenation. Moreover, ortho- or para-alkoxyarylketone is also highly reactive. Therefore, the experimental data presented in Figures 4 and 5 are not surprising results.

Reviewer #3 (Remarks to the Author):

The manuscript by Hall and Rygus describes a hemiboronic acid catalytic platform for the dual activation of alcohols. Based on their previous study of structure and properties of benzoxazaborine 2 (ref 19), the authors designed and prepared a closely related benzoxazaborinium salt 3. While the moderate Lewis acidity of 2 facilitates the nucleophilic phosphorylation of diols, the strong acidity of cationic catalyst 3 enables the generation of carbocation through electrophilic activation of alcohols. Hall's group has been making continued contributions to the boron chemistry. In the current work, the concept of achieving divergent catalytic transformations of common chemical entities through distinct activation modes by using structurally related catalysts is smart and inspiring.

It's also very important that the authors have performed extensive research to elucidate two distinct activation modes by the boron catalysts and the mechanism of the corresponding divergent catalytic reactions of alcohol substrates. The reported transformations, the monophosphorylation of diols and reductive deoxygenation of benzylic alcohols or ketones, even if not brand new, show significant improvements over the previous methods involving metal catalysts and popular boron catalyst $B(C_6F_5)_3$. The easy preparation of the organocatalyst and mild reaction conditions may attract interest for potential applications in organic synthesis.

This manuscript is very well written and the results are convincing. The Supporting Information is also clear and informative. In my opinion this work worth to be published in *Nature Communications* after addressing the following comments:

1. A more informative title should include "boron catalyst" as a keyword.
2. While the design concept of catalyst and activation mode is impressive, the significance of the reaction methodology part needs to be strengthened. The application of current methods to the later stage modification of complex molecules will make this work more appealing.
3. As a key intermediate, is it possible to isolate and structurally characterize (X-ray) borate 2-II?
4. In Fig 1a, "Y" should be defined to avoid misunderstanding.
5. It is interesting to know whether the phosphorylation of 1,3-diol works (through a six-membered ring borate intermediate).
6. A brief explanation of the regioselectivity of the monophosphorylation should be added.
7. In the legend of Fig. 2, "monophosphorylation" should be "monophosphorylation".

NCOMMS-22-47097 – Response to Reviewer Evaluations and Additional Modifications

Reviewer #1

This manuscript reports the exploration of benzoxazaborines for the catalytic activation of alcohols. This work is unique in that the same parent scaffold can be used for the activation of alcohols towards reaction with electrophiles and nucleophiles, which have very different requirements of the catalyst. The parent benzoxazaborine catalyst enhances the nucleophilicity of diols, which is exemplified by a mono-phosphorylation reaction. Formation of the N-methylated benzoxazaborinium analogue changes the nature of the catalyst, allowing activation of alcohols and ketones towards reduction into the corresponding alkane using alkylsilanes. Mechanistic control experiments are provided for both sets of reactions to provide supporting evidence for the differing roles of the catalysts in each process. The substrate scope and limitations are also discussed for both reaction classes. The manuscript is prepared to a very high standard and was a pleasure to read. The Supporting Information is detailed, the products well characterised, and the NMR spectra show sufficient purity.

While the individual reactions reported have some precedence in arylboronic / borinic acid catalysis, the breakthrough in this work is the development and demonstrated versatility of the benzoxazaborine scaffold for catalysis. The mechanistic insights shown ensures that there is great potential for further exploration of these interesting catalysts that I am sure will be of wide interest to the community. Publication in Nature Communications can therefore be highly recommended, essentially without correction.

RESPONSE: We thank the Reviewer for these constructive comments.

Minor Comment

SI Page 6 – The graph in Figure S1 is of low resolution compared with the other figures. It is still readable, but the authors may want to update.

RESPONSE: We thank the Reviewer for noticing this error. The graph has been replaced in higher resolution.

Reviewer #2

Hall et al. developed boronic acid catalyst for monophosphorylation of vicinal diols and boronic acid catalyst for reductive deoxygenation of alcohols and ketones. These two boronic acid catalysts were designed based on a common benzoxazirine scaffold. However, these simple two reactions have been already known. The novelty and substrate scope are not acceptable as a paper in Nat. Commun. Monophosphorylation of vicinal diols: In general, it is well known that primary hydroxy group more reactive than secondary hydroxy group. In particular, secondary hydroxy group of 1-aryl-1,2-ethanediol is much less reactive than primary hydroxy group. The regioselectivity should be shown in Figure 2. The substrate scope is quite limited. Reductive deoxygenation of alcohols and ketones: In general, diarylmethanol and diarylketone are highly reactive for the reductive deoxygenation. Moreover, ortho- or para-alkoxyarylketone is also highly reactive. Therefore, the experimental data presented in Figures 4 and 5 are not surprising results.

RESPONSE: We thank the Reviewer for their evaluation. We partially understand the point of view of the Reviewer regarding the reactions exemplified in the manuscript. It appears, however, that two important points may have been overlooked. While it is correct that electrophilic and nucleophilic activation of alcohols can be achieved with boronic/borinic acids, to the best of our knowledge diol phosphorylation and alcohol/ketone deoxygenation have never been achieved with this type of mild catalysis, which presents distinct advantages. Moreover, those particular reactions were not the nascent point of this study. The primary intent was not to invent brand new reactions, and the main advance is best stated by the comments of Reviewer #1 and #3:

"the breakthrough in this work is the development and demonstrated versatility of the benzoxazaborine scaffold for catalysis."

"In the current work, the concept of achieving divergent catalytic transformations of common chemical entities through distinct activation modes by using structurally related catalysts is smart and inspiring."

Figure 2: The regioselectivity was addressed in the figure legend.

Reviewer #3

The manuscript by Hall and Rygus describes a hemiboronic acid catalytic platform for the dual activation of alcohols. Based on their previous study of structure and properties of benzoxazaborine **2** (ref 19), the authors designed and prepared a closely related benzoxazaborinium salt **3**. While the moderate Lewis acidity of **2** facilitates the nucleophilic phosphorylation of diols, the strong acidity of cationic catalyst **3** enables the generation of carbocation through electrophilic activation of alcohols. Hall's group has been making continued contributions to the boron chemistry. In the current work, the concept of achieving divergent catalytic transformations of common chemical entities through distinct activation modes by using structurally related catalysts is smart and inspiring.

It's also very important that the authors have performed extensive research to elucidate two distinct activation modes by the boron catalysts and the mechanism of the corresponding divergent catalytic reactions of alcohol substrates. The reported transformations, the monophosphorylation of diols and reductive deoxygenation of benzylic alcohols or ketones, even if not brand new, show significant improvements over the previous methods involving metal catalysts and popular boron catalyst $B(C_6F_5)_3$. The easy preparation of the organocatalyst and mild reaction conditions may attract interest for potential applications in organic synthesis.

This manuscript is very well written and the results are convincing. The Supporting Information is also clear and informative. In my opinion this work worth to be published in *Nature Communications* after addressing the following comments:

RESPONSE: We thank the Reviewer for these constructive comments.

1. A more informative title should include “boron catalyst” as a keyword.

RESPONSE: This is an excellent suggestion to make the title more generally accessible. The title has been modified as suggested to the following: *Direct nucleophilic and electrophilic activation of alcohols using a unified boron-based organocatalyst scaffold.*

2. While the design concept of catalyst and activation mode is impressive, the significance of the reaction methodology part needs to be strengthened. The application of current methods to the later stage modification of complex molecules will make this work more appealing.

RESPONSE: We thank the Reviewer for this suggestion. The late-stage functionalization of functionalized benzophenone- or benzhydryl-containing bioactive compounds cloridarol, fenofibrate and ketoprofen are demonstrated in the synthesis of products **7t**, **7aj** and **7ak** respectively. Additionally, the use of the deoxygenation protocol to produce an intermediate in the synthesis of diabetes treatment sotagliflozin is demonstrated in a new example, **7ai**, which proceeds in improved yield relative to the literature precedent employing harsh conditions such as excess boron trifluoride diethyl etherate. Application of this methodology towards the deoxygenation of terfenadine and haloperidol were unsuccessful, in line with the failure of other substrates containing nitrogen heterocycles as described in the Supporting Information.

3. As a key intermediate, is it possible to isolate and structurally characterize (X-ray) borate **2-II**?

RESPONSE: We thank the reviewer for this insightful suggestion. Our studies indicate that under the reaction conditions, borate **2-II** exists as a mixture of two diastereomers (due to tetracoordinate boron being stereogenic). When using DIPEA as a base (as is employed in the optimized monophosphorylation reaction conditions), these two diastereomers appear to interconvert at a moderate rate relative to the NMR timescale, resulting in extremely broad resonances for protons corresponding to either the diol or boron heterocycle components. Conversely, the use of tetrabutylammonium hydroxide as a base appears to generate two diastereomers that are slow to equilibrate on the NMR timescale, and two distinct sets of resonances can be observed in approximately a 60:40 ratio. A statement addressing this observation has been added in the text of the main paper, and the corresponding ¹H NMR figures are included in the supporting information as Figures S5 and S6.

We believe that the rate of equilibration may be influenced by the Brønsted acidity of the conjugate acid from the base used in borate formation: a protonated trialkylammonium species is several orders of magnitude more acidic than H₂O, which may result in more rapid equilibration of the two diastereomers via a trivalent boron intermediate when DIPEA is used as a base relative to the tetrabutylammonium hydroxide. Accordingly, crystallization of **2-II** is likely not possible due to the mixture of two diastereomers. Further studies will be carried out in our laboratory to understand the effect of this equilibrium on nucleophilic activation of 1,2-diols.

4. In Fig 1a, “Y” should be defined to avoid misunderstanding.

RESPONSE: Figure 1a has been amended for additional clarity to differentiate the products of electrophilic and nucleophilic activation.

5. It is interesting to know whether the phosphorylation of 1,3-diol works (through a six-membered ring borate intermediate).

RESPONSE: We thank the Reviewer for this suggestion. Catalytic monophosphorylation of 1,3-diols was found to proceed much more slowly than vicinal diols. Using both 2-phenyl-1,3-propanediol or 1-phenyl-1,3-propanediol, approximately 60% conversion was observed after 24 h by crude NMR under the conditions employed for 1,2-diols. Similar results can be obtained in the absence of catalyst by heating the reaction to 60 degrees C. As described in the Supporting Information, catalytic monophosphorylation of 1,2-diols was sufficiently fast to provide significant and synthetically useful rate acceleration relative to the uncatalyzed background reaction, which does not seem to be the case for 1,3-diols. These results are consistent with our group's previous report on the benzylation of 1,3-diols using hemiboronic acid catalysis, which was conducted for 24 hours (reference 16 of the manuscript).

6. A brief explanation of the regioselectivity of the monophosphorylation should be added.

RESPONSE: The regioselectivity observed in our system is in line with that observed previously by Taylor using borinic acid catalysts, in which primary hydroxy groups are functionalized preferentially over secondary hydroxy groups (see *J. Am. Chem. Soc.* **2012**, *134*, 8260–8267, reference 14 in the document).

The following text was modified accordingly: ". In line with a sterically preferred attack of the least hindered oxygen atom of borate complex **2-II**, phosphorylation occurred with complete regioselectivity for the primary alcohol in all cases."

7. In the legend of Fig. 2, "monophorphylation" should be "monophosphorylation".

RESPONSE: We thank the Reviewer for noticing this mistake. It has been corrected.

Additional Changes

- graphic abstract has been slightly modified
- small changes in Figure 1 to make the depiction of charges more clear
- substrate scope of ketone reduction (Figure 5) has been renumbered and reordered
- additional NMR experiment added to probe the possibility of borohydride formation using TMDSO (Supplementary Figure 28)
- Methods and Data availability sections added at the end of the manuscript
- the same reference was erroneously given two different numbers in the Supplementary Information. This has been corrected

REVIEWERS' COMMENTS

Reviewer #3 (Remarks to the Author):

The response letter has fully addressed my comments and concerns. It's impressive that the authors have made extra effort to improve their manuscript. I appreciate the authors for their professional work. Therefore, I recommend the current manuscript to be published in Nature Communications.